# Application of Heat-Enhancement for Improving the Sensitivity of Quartz Crystal Microbalance

**DOI:** 10.3390/bios12080643

**Published:** 2022-08-15

**Authors:** Chenglong Song, Zhihao Ma, Chenglong Li, Hongxing Zhang, Zhiqiang Zhu, Jie Wang

**Affiliations:** 1Institute for Advanced Materials, School of Material Science and Engineering, Jiangsu University, Zhenjiang 212013, China; 2Department of Medical Science and Technology, Suzhou Chien-Shiung Institute of Technology, Suzhou 215411, China

**Keywords:** quartz crystal microbalance, resonant frequency, single nucleotide polymorphism (SNP)

## Abstract

The use of quartz crystal microbalance in trace mass detection is restricted by unsatisfactory sensitivity, especially in damping media, due to the worsening of the quality factor of the damping resonator. The enhancement of the sensor performance could be realized by increasing the innate resonant frequency of quartz oscillators. Herein, increased working temperature of QCM systems was proved to bring an enhancement of the original resonant frequency. In addition, the measurement of ion osmotic pressure, single layer formation and single nucleotide polymorphism (SNP) at different temperatures demonstrated that an increased working temperature could enhance the sensitivity and accuracy, suggesting a potential application in a series of trace detections.

## 1. Introduction

Quartz crystal microbalance (QCM) is a well-known sensor based on piezoelectric quartz crystal for detecting trace changes in mass. QCM platform has many advantages such as being label-free, making it a superior tool in various fields such as online monitoring of industrial vacuum coatings and detection of atmospheric and environmental pollutants [1]. In recent years, QCM has been further applied in liquids based on biomolecular interactions, adsorption [2,3,4] and DNA hybridization [5,6,7,8,9]. Despite its numerous applications, QCM still suffers from its relatively limited sensitivity. For example, it is a challenge to apply a conventional QCM to study the affinity of target proteins at low concentrations [10]. QCM is far less popular compared with surface plasma resonance (SPR) as a biosensor due to its limited sensitivity [11].

The traditional QCM is constructed upon the piezoelectric quartz crystal. According to Sauerbrey’s equation (Sauerbrey, 1959), the resonant frequency of quartz crystal could be precisely determined and small changes of mass (ng/cm^2^) in vacuum and in air could be calculated. However, the resolution of QCM decreases due to the damping effect of liquid. According to the literature, the resolution of QCM was enhanced with the improvement of the working frequency [12]. As shown in Equation (1), for any given frequency change (Δf), mass change (Δm) is proportional to the square of resonant frequency (*f*_0_) of a QCM resonator. Increased *f*_0_ will improve the sensitivity and resolution of QCM.
(1)$$C=2f_{0}^{2}/(A\sqrt{\rho \mu })$$
where *C* is a crystal-dependent sensitivity factor that includes the fundamental resonant frequency (*f*_0_), the surface area (*A*) and two constants of quartz crystal (*μ*: shear modulus and *ρ*: quartz density) [13].

Therefore, to enhance the sensitivity of QCM sensors, it is imperative to develop a high-frequency resonator, especially for highly sensitive applications in liquid surroundings. The traditional way of fabricating a QCM chip requires a series of challenging fabrication processes to decrease the thickness of resonators, e.g., the ultra-thin quartz wafer should be controlled below 8.3 μm for up to 200 MHz quartz resonators. However, the resonators turn more vulnerable with the decreasing thickness. The application of high-frequency resonators has been hindered by the fabrication techniques and vulnerability of ultra-thin quartz crystal wafers.

According to previous studies, in addition to the quality and structure of quartz wafer, other factors also affect QCM signals and one of the factors is temperature. Therefore, inadequate temperature stability in the measurement process will introduce temperature-related pseudo-signals into the data, and ultimately it will be impossible to distinguish how much of the measurement signal is derived from temperature changes or surface interactions under study. The environment temperature shift could affect QCM frequency signals, since commonly employed commonly-employed AT-cut quartz crystals are still sensitive to the change of temperature (AT-cut: the quartz crystals are cut at an angle of 35°15′ from the Z-axis.), especially in the vicinity of room temperature [14]. Herein, in most applications, the temperature fluctuation is regarded as an adverse factor influencing the accuracy of measurement and is always strictly monitored and minimized. In the past, various methods of thermal compensation were investigated for eliminating the temperature fluctuations of a quartz crystal resonator [14,15,16,17,18]. To the best of our knowledge, no research has been reported to positively utilize the temperature dependence of QCM frequency to improve QCM sensitivity.

In this study, we applied the temperature dependence of QCM to enhance its resonant frequency and detection performance. A second excitation of heating superimposed on the traditional electric excitation on quartz resonators was utilized to attain a higher fundamental frequency and improve QCM sensitivity and accuracy in a series of measurements, including ion concentrations, and assembly of small molecule monolayer and single nucleotide polymorphism (SNP) detection.

## 2. Methods

In this research, all QCM measurements were performed on a Q-Sense E4 apparatus (Biolin Scientific, Stockholm, Sweden). In the measurement, the sample injection was controlled by a peristaltic pump at a speed of 50 μL/min. The QCM chips with a thickness of 0.33 mm were purchased from Biolin Scientific Ltd (Biolin Scientific, Stockholm, Sweden). The resonant frequency of QCM chips is 5 MHz. All aqueous solutions were prepared with twice-deionized water.

### 2.1. Atomic Force Microscopy (AFM)

The QCM chips were incubated separately with DNA samples including “PM”, “1 M” and “control” which were target DNA samples consisting of different DNA sequences and had different affinity to hybridize with probe DNA on the QCM chips. All AFM measurements were accomplished on an equipment (Multimode 8, Bruker, USA) in tapping mode and with a silicon cantilever (OMCL-AC160TS-R3, Olympus). The analyses of AFM data were accomplished by the Gwyddion software versions 2.54.

### 2.2. The Determination of Ion Concentration of Sodium Chloride and Phosphate-Buffered Saline (PBS)

A sodium chloride solution (0.16 mM) was prepared and consecutively diluted by 25-fold and then injected into the QCM chamber. The tiny ion concentration shifts were monitored online by QCM at different temperatures, which were controlled by the temperature unit inbuilt inside QCM equipment.

1X PBS buffer was purchased from GE Healhcare Lifescience (Hyclone, GE, USA). 1X PBS buffer of pH 7.0~7.2, containing 6.7 mM PO_4_ ion was diluted 5-fold and consecutively injected into the QCM chamber. The mixed ion concentration shifts were monitored online by QCM at different temperatures.

### 2.3. The Monitoring of Self-Assembly of 11-Mercaptoundecanoic Acid (MUA)

The MUA reagent was purchased from Sigma-Aldrich (Steinheim, Germany). The MUA powder was dissolved in ethanol and diluted to 1 mg/mL and then injected into the QCM chamber. The self-assembly of MUA molecules was monitored online by QCM at different temperatures.

### 2.4. Single Nucleotide Polymorphism (SNP) Detection

The SNP detection was performed according to the published works with some modifications [19]. The DNA sequence of “probe”, “1 M”, ”PM” and ”Control” is provided in Table 1. Immobilization of probes: 1 µM probe solution abbreviated as “Probe” was prepared with 1 M KH_2_PO_4_ solution. In the beginning, 1 mM Mercaptohexanol (MCH) solution was prepared with ethanol. DNA Hybridization solutions contained 1 M NaCl in TE buffer (10 mM Tris buffer, 1 mM EDTA, pH 7.6). To achieve a probe density of 3.0 × 10^12^ probes/cm^2^, the gold surface of the QCM chip was exposed to a 1 µM solution of a thiolated DNA probe for ~2 h online. After probe immobilization, 1 mM mercaptohexanol (MCH) solution was pumped into the QCM chamber and incubated for 2 h to avoid nonspecific adhesion on the chip surface. The working temperature was then increased to 62 °C before measurements and the chips were rinsed then for 10 min online.

Measurement of DNA hybridization: For SNP hybridization, the hybridization solution was pumped into the QCM chamber to set up a steady baseline. Then 1 µM target DNA samples (including “PM”, “1 M” and “Control”) were consecutively pumped into the QCM chamber and incubated for 8 h at 37 °C online. Before the injection of each sample, the QCM chips were rinsed at 62 °C for 10 min online.

## 3. Results and Discussion

### 3.1. Temperature Dependency of the Resonant Frequency and Energy Dissipation of QCM

The resonant frequency and energy dissipation were assayed at 25 °C, 37 °C and 55 °C both in air and in liquid. In this study, the resonant frequency shift was calculated by the frequency signals at third overtone (∆F_3_) which is further normalized into ∆F_3_/3 to avoid the interference of different overtones of QCM. The energy dissipation curve was recorded by the QCM apparatus (Biolin, Qsense E4). As shown in Figure 1a,c, the temperature shifts produced constant and reproducible frequency signals both in air and water. The frequency shifts due to the temperature shifts were much larger in water than in air which may be attributed to the enhanced efficiency of heat exchange in water. Linear relationships were observed between the frequency shift and temperature (Figure 1e,f). As shown in Figure 1b, the energy dissipation in the air is too small to indicate temperature fluctuations. By contrast, the energy dissipation was enhanced in water, as shown in Figure 1d. The linear relationship between energy dissipation and the temperature was shown in Figure 1f. The above results confirmed that the temperature could enhance the resonant frequency of QCM in air and water and the temperature shift could also induce a discernible energy dissipation in water, which was in accord with previous reports [15,16,17,18]. As mentioned above in the introduction section, a higher resonant frequency could improve the sensitivity of QCM. Herein, the temperature dependency of QCM frequency was further investigated on the performance of QCM in a series of measurements.

### 3.2. The Temperature Dependency of QCM Applied for the Ion Detection

QCM was used to monitor the concentration changes of ions online by QCM at 25 °C, 37 °C and 55 °C, respectively. As shown in Figure 2a,b, the concentration changes of sodium chloride solutions illicit perceivable frequency and energy dissipation signals. Furthermore, a higher working temperature improved the sensitivity of QCM. The QCM frequency signals at 25 °C were much weaker than its counterpart at 55 °C. At 25 °C the transfer from sample transfer of “a solution” to “f solution” induced a 4.5 Hz frequency shift, in comparison to the 8.5 Hz at 55 °C. The shift of frequency and energy dissipation signal cannot well distinguish the ion concentration changes from “c solution” to “f solution” at 25 °C and 37 °C, but the signals gained at 55 °C could clearly indicate changes in concentration of NaCl.

To investigate temperature dependency for detection of a mixture buffer of anions and cations, a PBS buffer purchased from GE Healcare Lifescience containing 6.7 mM PO_4_ ion and Na, K, and Cl ions was also tested. A series of PBS 25X-dilutions were prepared and tested by QCM the same as sodium chloride solution. As shown in Figure 2c,d, both the frequency and energy dissipation signals were obtained. At 55 °C, QCM was more sensitive to the changes in concentration of PO_4_, Na, K, and Cl ions, which is clearly shown in Figure 2e,f. All experimental results suggested a higher working temperature enhanced the sensitivity of QCM for ions detection.

### 3.3. The Temperature Enhancement of QCM Applied for MUA SAMs Monitoring

The temperature enhancement of QCM was further investigated by monitoring the self-assembly of a single layer of 11-Mercaptoundecanoic acid (MUA). After a setting-up of the QCM baseline of ethanol, 1 mg/mL MUA in ethanol solution was injected into the QCM chamber and formed a self-assembled monolayer (SAM) on the gold surface through an Au-S bond. QCM was utilized to monitor the self-assembly process online at different temperatures. As shown in Figure 3a, after an incubation of 720 min at 25 °C, the self-assembly of MUA induced a frequency shift of 3.2 Hz, compared with a frequency shift of 16 Hz measured at 37 °C. The energy dissipation presented a similar result that high working temperature induced stronger signals (Figure 3b). The energy dissipation at 37 °C was twofold that of 25 °C. The frequency and energy dissipation signals were summarized in Figure 3c. The conclusion was in accordance with Figure 2, indicating a temperature enhancement of QCM sensitivity.

### 3.4. The Temperature Enhancement of QCM for the Single Nucleotide Polymorphism (SNP) Detection

To test the temperature enhancement of QCM on the DNA detection, an SNP experiment was prepared as previously described [19]. A layer of DNA probe was pre-immobilized onto the QCM chip. The above chips were then incubated overnight in 1 mM MCH solution to form an MCH layer. Finally, the different target DNAs including “PM”, “1 M” and “Control” DNAs were consecutively pumped into the QCM chamber. DNA hybridization at 20 °C was monitored as shown in Figure 4a. The DNA hybridization between the Probe and “PM” which is completely complementary to the Probe strand elicited a frequency shift of 11 Hz after an incubation of 7 h and the signals could not be distinguished from the other two samples (“1 M” and “Control”), which were a single base-pair mismatch and noncomplementary to “probe” strand, respectively. At 40 °C the QCM signal was enhanced to discern the target DNAs “1 M” and “Control”. The “PM” induced a frequency shift of 24 Hz, which was almost two-fold that of 25 °C.

The AFM was used to characterize the DNA hybridization of different samples on the gold surface of chips. After incubation with target DNA (including “PM”, “1 M” and “Control”) separately, the QCM chips were rinsed and characterized by AFM. In the AFM measurement, DNA hybridization formed a stake-like structure with a thickness of 7.3 nm, which was quite close to the theory calculation by double DNA structure with 25 base-pair and 0.34 nm distance for neighbouring base pairs. As shown in Figure 4e–g, the stake-like structure formed by DNA hybridization was evidently represented by the dot with a lighter colour in AFM mages. A more stake-like structure was found on chips incubated with a “PM” solution than those incubated with “PM”, “Control” and “1 M” samples. The roughness of AFM images further showed the roughness of AFM images gradually increased in “PM” sample compared with “Control”, “1 M” and “PM” samples.

### 3.5. The Mechanism of Temperature Enhancement of QCM

The resonant frequency of quartz resonator immersed in liquid is sensitive to the following factors: (1) mass changes on the electrode surface; (2) the properties (i.e., density and viscosity) of the liquid in contact with the quartz crystal; (3) the temperature and pressure of the quartz resonator [20]. In most thin-film deposition in liquid, the mechanical stresses and a roughness effect of the electrode surface of the quartz resonator could be neglected because the quartz resonator is always immersed in a liquid at a constant depth. Summarily, the resonant frequency (Δf) can be written as [14,15]:(2)$$\Delta f=\Delta f_{m}+\Delta f_{\mu \eta }+\Delta f_{T}$$
where the subscripts *m*, *μη* and *T* refer to the mass change on the surface, viscosity change of liquid and temperature effects of the quartz resonators, respectively.

Further, the mass change induced frequency shift $\Delta f_{m}$ could be calculated according to the well-known Sauerbrey equation and represented as Equation (3). The density and viscosity-induced frequency shift $\Delta f_{\mu \eta }$ could be calculated according to Equation (4) [14,15]. The temperature-induced frequency shift $\Delta f_{T}$ could be calculated according to Equation (5) [14,15].
(3)$$\Delta f_{m}=-2.26\times 10^{-6}\frac{f_{0}^{2}}{A}\Delta m$$
(4)$$\Delta f_{\mu \eta }=-f_{0}^{3/2}\sqrt{\frac{\rho_{l}\mu_{l}}{\pi \rho_{q}\mu_{q}}})$$
(5)$$\Delta f_{T}=a_{1}f_{0}\left(T-T_{0}\right)+a_{2}f_{0}{\left(T-T_{0}\right)}^{2}+a_{3}f_{0}{\left(T-T_{0}\right)}^{3}+\ldots$$
where *T*_0_ is a reference temperature; *f*_0_ is the resonant frequency at *T*_0_; *a*_1_, *a*_2_, and *a*_3_ are three constants. $\mu_{l}$ and $\rho_{l}$ are the solution viscosity and solution density, respectively. Both of them are dependent variables of the temperature. $\mu_{q}$ and $\rho_{q}$ are the quartz viscosity and quartz density, respectively.

For a series of applications of QCM in liquid at a given temperature, such as the aforementioned QCM applications in detections of ion concentration, SAM assembly and DNA mismatches, the frequency shift should be determined by two variables: $\Delta f_{m}$ and $\Delta f_{\mu \eta }$. $\Delta f_{\mu \eta }$ is a negative value and $\Delta f_{m}$ sometimes is a positive value when the mass on the QCM chips surface decreases (Δm is a negative value). For example, in the detection of decreasing concentrations of NaCl solutions, the $\Delta f_{m}$ is a positive value because Δm is a negative value. $\Delta f_{\mu \eta }$ is also a positive value (its absolute value is decreasing) because the density and viscosity of NaCl solution deceased with the reduced concentration, in accord with the rising tendency of the QCM frequency curve in Figure 2.

For a series of applications of QCM in liquid at an elevated temperature, the temperature change of quartz resonator-induced frequency shift ($\Delta f_{T}$) should be accounted for. Due to insufficient data existing for the determination of the temperature derivatives of the piezoelectric and dielectric constants, the temperature effect on electric field and piezoelectricity has not been well-analyzed in the past [21]. According to several reports in this field, the temperature effect on the elastic constants of quartz of AT-cut crystals is represented by a power series expanded up to the third order in the temperature ranging from −60 to +100, as shown in Equation (5) [14,22].

According to Figure 1, a rising tendency of frequency curve was observed with temperature elevated from 25 °C to 55 °C in air, which was also consistent with previous study [15]. Our experiment data and previous literatures support that higher temperatures benefit the elevation of the resonant frequency of quartz resonators. According to Equation (1), a higher resonant frequency could finally improve the sensitivity of QCM. This theoretical deduction has been demonstrated in a series of measurements in this study.

α-Quartz is the most commonly used piezoelectric material at present. The piezoelectric properties of α-quartz-based resonators are found to degrade above 300 °C which is well below the transition at 573 °C to the β phase of quartz crystal [23]. In this study, the applied working temperature varied from 25 °C to 50 °C, which was limited by the temperature control unit of QCM equipment. With a higher temperature, it is expected that the performance of quartz-based resonators would be further improved.

## 4. Conclusions

In summary, the measurements of ions, SAM and DNA mismatches at different temperatures revealed that the elevated temperature brought an enhancement of resonant frequency and energy dissipation, which eventually improved the detection sensitivity. This strategy opens up a new revenue to positively utilize the temperature control to increase the quartz resonant frequency and promote its applications in some appropriate fields where high temperature is not a limitation.

## Figures and Tables

**Figure 1 biosensors-12-00643-f001:**
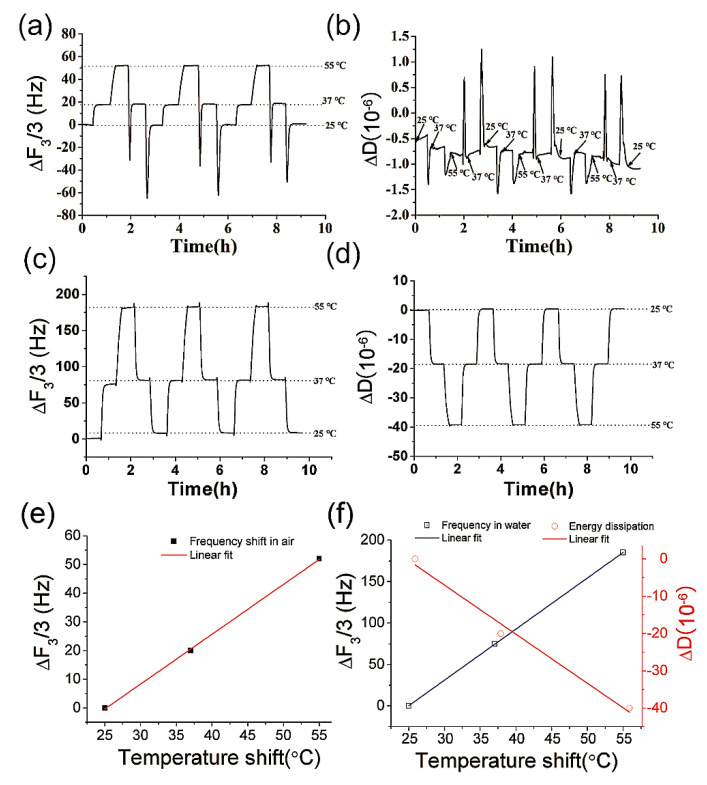
The temperature changes induced frequency and energy dissipation shifts of QCM in air and water. (**a**) the frequency shift in air; (**b**) energy dissipation shift in air; (**c**) the frequency shift in water; (**d**) the energy dissipation shift in water; (**e**) the linear fitting for the temperature and QCM frequency in the air; (**f**) the linear fitting for the temperature and QCM signals in water.

**Figure 2 biosensors-12-00643-f002:**
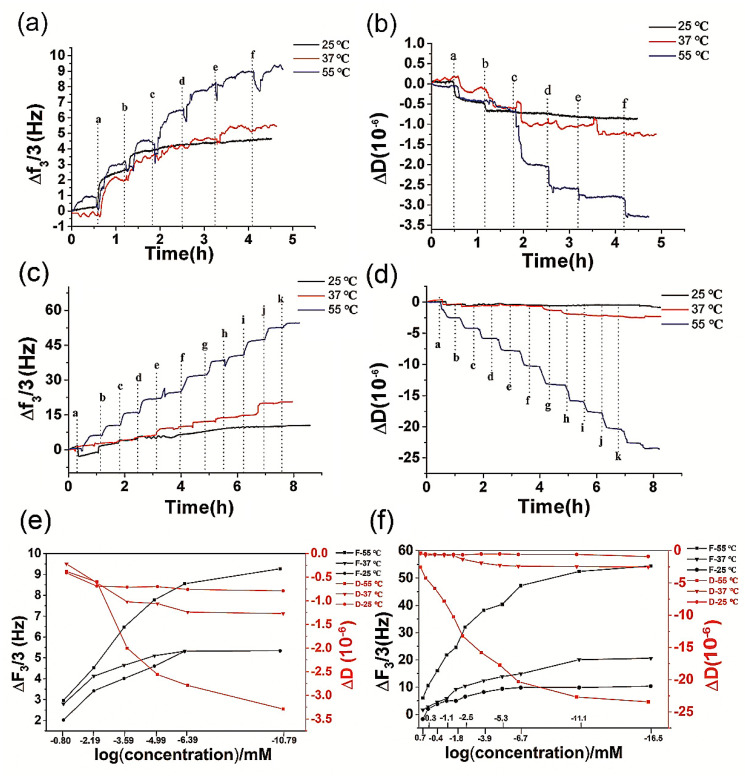
QCM was utilized to monitor ion concentrations of different solutions at 25 °C, 37 °C and 55 °C; (**a**) the frequency shifts for sodium chloride solutions. From “a” to “f” samples, the sodium chloride solutions are of 0.16 mM, 0.0064 mM, 0.000256 mM, 0.00001024 mM, 0.0000004096 mM, 0.000000016384 mM, respectively; (**b**) the energy dissipation for sodium chloride solutions; (**c**) the frequency shifts for PBS buffers. From “a” to “j” samples, 1X PBS buffer was consecutively diluted by 5-fold with the final sodium concentrations of 0.4 mM, 0.08 mM, 0.016 mM, 0.0032 mM, 0.00064 mM, 0.000128 mM, 0.0000256 mM, 0.0000512 mM, 0.000001024 mM, 0.0000002048 mM, respectively. “k” is pure water; (**d**) energy dissipation for PBS buffers; (**e**) the frequency shifts for sodium chloride solutions with different concentrations at different temperatures; (**f**) the frequency shifts for PBS buffer with different ion concentrations at different temperatures.

**Figure 3 biosensors-12-00643-f003:**
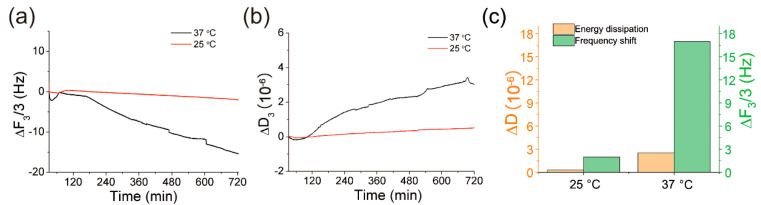
QCM was utilized to monitor the self-assembly of 11-Mercaptoundecanoic acid (MUA) on the gold surface of QCM chips at 25 °C and 37 °C. (**a**) the frequency; (**b**) energy dissipation; (**c**) the summary of QCM signals.

**Figure 4 biosensors-12-00643-f004:**
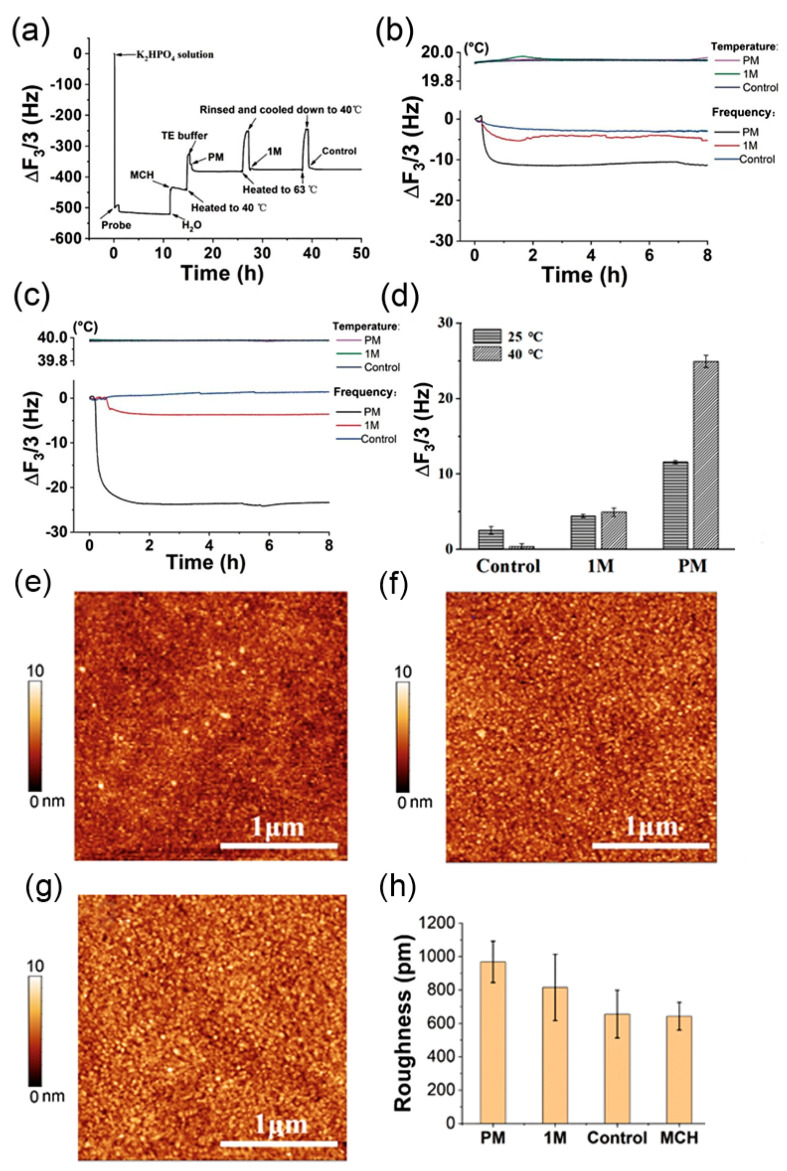
QCM was utilized to measure SNP at different temperatures. (**a**) the frequency of the entire QCM experiment; (**b**) frequency shift for SNP measurement at 20 °C; (**c**) frequency shift for SNP measurement at 40 °C; (**d**) the comparison of QCM frequency shifts at 25 °C and 40 °C; (**e**) AFM image of “Control” sample; (**f**) AFM image of “1 M” sample; (**g**) AFM image of “PM” sample; (**h**) roughness of AFM images of different samples. The scale bar of the AFM image is 1 μm.

**Table 1 biosensors-12-00643-t001:** DNA sequences.

Samples	Sequence
probe	AGATCAGTGCGTCTGTACTAGCACA
PM	TGTGCTAGTACAGACGCACTGATCT
1 M	TGTGCTAGTACAGACACACTGATCT
Control	AGATCAGTGCGTCTGTACTAGCACA

probe was modified with 5′ Thiol Modifier C6-S-S.

## Data Availability

Not applicable.

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
