# Peer review of "Application of Heat-Enhancement for Improving the Sensitivity of Quartz Crystal Microbalance"

_biosensors, 2022, doi:10.3390/bios12080643_

Round 1

Reviewer 1 Report

line 41: it might be better to put the values in SI rather than in CGS 

Figure 1 : add the units on the x-axis of the graph (b)

Author Response

Reviewer 1:

Comments and Suggestions for Authors

Q1: line 41: it might be better to put the values in SI rather than in CGS

R1: Yes, values are usually put in SI rather than in CGS. However, in QCM-related experiments, “cm-2” is usually used as the unit (Ieee Sens J 2005, 5 (6), 1251-1257; Chemosensors 2022, 10, 172.). Here, we have removed the values of shear modulus (μ) and quartz density (ρ) in the manuscript.

Q2: Figure 1 : add the units on the x-axis of the graph (b)

R2: Thank you for your suggestion. Figure 1 has been revised.

Reviewer 2 Report

Ref: biosensors- 1788430

Title of the manuscript: Repurposing Heat-enhancement for Reactivating Quartz Crystal Microbalance for Sensitivity and Accuracy.”

 In this paper, Zhihao Ma et al. involved the use of a quartz crystal microbalance for trace mass detection in air and water media at different temperatures to enhance sensitivity and accuracy. The article looks nice and interesting but requires extensive English editing. Proper spacing between the units and numerals must be carefully checked. More references should be incorporated especially in Section 3, if possible, to support the results. Figure captions should be rechecked for proper formatting. Insert a table for the comparison of the findings and literature reports.

Author Response

Reviewer 2:

Comments and Suggestions for Authors

Ref: biosensors- 1788430

In this paper, Zhihao Ma et al. involved the use of a quartz crystal microbalance for trace mass detection in air and water media at different temperatures to enhance sensitivity and accuracy. The article looks nice and interesting but requires extensive English editing.

Q1: Proper spacing between the units and numerals must be carefully checked.

R1: Thank you for your suggestion. We have checked the manuscript and added spaces between units and values.

Q2: More references should be incorporated especially in Section 3, if possible, to support the results.

R2: Thanks reviewer for suggestions. Five relevant publications have been included in the modified manuscripts.

Q3: Figure captions should be rechecked for proper formatting.

R3: Thank you for your suggestion. We have checked and revised figure captions. 

Q4: Insert a table for the comparison of the findings and literature reports.

R4: Because previous studies and this study applied different QCM apparatus and chips, especially the QCM chips of different resonant frequency, these experiment conditions would greatly influence the frequency and energy dissipation values. It is unappropriate to contrast the data directly on different platforms and QCM chips.

In QCM measurement, we usually arrange a contrast among different sample groups to test different parameters effect based on the uniform platform. In this study, we set up different temperature groups to clarify temperature effect, which is also partially confirmed in other publications and we have added the comparison in the discussion of mechanism.

Reviewer 3 Report

The manuscript "biosensors-1788430" entitled "Repurposing Heat-enhancement for Reactivating Quartz Crystal Microbalance for Sensitivity and Accuracy" deals with the effect of  temperature of QCM systems on enhancing the sensitivity and accuracy of the measurements. The followings comments need to be addressed by the authors:

1. The title and the scope of the work do not match. When I read the title I expected to see some elements of design to improve QCM measurements, not just studying the temperature effect. The authors must revise the title to make it more relevant to the presented work.

2. Why there is a reference (ref. 15) cited for the title of section 2.4? Has the entire section 2.4 been extracted from ref. 15?    

3.The results are not contrasted and compared to relevant literature

4. What is the purpose of the AFM images? Can the authors present more relevant images (e.g., images taken under the same QCM experimental conditions including the composition of the medium)?

Author Response

Reviewer 3:

Comments and Suggestions for Authors

The manuscript "biosensors-1788430" entitled "Repurposing Heat-enhancement for Reactivating Quartz Crystal Microbalance for Sensitivity and Accuracy" deals with the effect of  temperature of QCM systems on enhancing the sensitivity and accuracy of the measurements. The followings comments need to be addressed by the authors:

Q1: The title and the scope of the work do not match. When I read the title I expected to see some elements of design to improve QCM measurements, not just studying the temperature effect. The authors must revise the title to make it more relevant to the presented work.

R1: Thank you for your suggestion. The title has been revised as “Application of Heat-enhancement for Improving the Sensitivity of Quartz Crystal Microbalance”.

Q2: Why there is a reference (ref. 15) cited for the title of section 2.4? Has the entire section 2.4 been extracted from ref. 15?    

R2: It is our mistake. We removed the reference cited for the title, and added “The SNP detection was performed according to the published works with some modification15.” in the beginning of the section.

Q3. The results are not contrasted and compared to relevant literature

R3: Because previous studies and this study applied different QCM apparatus and chips, especially the QCM chips of different resonant frequency, these experiment conditions would greatly influence the frequency and energy dissipation values. It is unappropriate to contrast the data directly on different platforms and QCM chips.

In QCM measurement, we usually arrange a contrast among different sample groups to test different parameters effect based on the uniform platform. In this study, we set up different temperature groups to clarify temperature effect, which is also partially confirmed in other publications and we have added the comparison in the discussion of mechanism.

Q4: What is the purpose of the AFM images? Can the authors present more relevant images (e.g., images taken under the same QCM experimental conditions including the composition of the medium)?

R4: AFM images were used to characterize the QCM chips before and after DNA hybridization, which could be compared with QCM data. The AFM mages in Figure 4e-g were analyzed by height distribution which presented DNA hybridization-formed stake-like dots with lighter color in AFM mages for Control, 1M to PM samples. This result confirmed DNA hybridization was enhanced in PM compared with Control and 1M samples as shown in Figure 4 e-h. The roughness of AFM images was further analyzed and compared, the AFM images for PM sample presented the roughest, demonstrating DNA hybridization was enhanced in “PM” (completely complementary) samples.

Reviewer 4 Report

The paper describes a simple method of altering temperature to increase the QCM sensitivity. Some doubts and suggestions are below:

1. Section 2.4: It is mentioned that hybridization solution contained target DNA. Then the authors have added hybridization solution followed by target DNA. Was target DNA added twice? 

2. Table 1 heading is misguiding. Target cannot include probe. 

3. The parameters in the y axis of Figures 1-4 have never been introduced. What are they and how they have been calculated? Why is frequency and energy expressed in these terms?

4. Have the authors considered other factors that change with temperature - such as solubility, Brownian motion etc?

5. Figure 2 illustrates change in signals with time and temperature. How is the concentration (of ions etc) factored in the results? Also, there is no mention of what were the changes in concentrations, or what range of ion concentrations were used. 

6. The paper title is unnecessarily complicated. What has been 'repurposed'? Is 'reactivating' the correct term?

7. A mechanism study would have helped strengthen the study. Have the authors measured the change in elastic constant, electric field and piezoelectricity with temperature?

There are numerous grammatical and syntactical errors in the text. A few examples are:

a. Line 15, 66: series

b. Line 21-22: Repetition of concept: 'well-known mass sensor' , 'for detecting trace changes in mass'

c. Line 23: what is 'real-time'? It is label-free rather than 'non-label'

d. Line 26: what of 'protein'. Repetition of 'such as'

e. Lines 47-48: repetition of 'thickness', 'thin/thinner', misuse of word 'thinner'

f. Line 57: what is meant by 'even though'. Expand AT

g. Line 73: move 'thickness of 0.33mm' in right position in sentence

h. Line 76: Expand abbreviations

i. Line 83, 87: 'The' is only used for a unique object. Buffers are not so. 

j. Line 85 and later: what is meant by 'QCM online'?

k. Line 92, 97: When sentences start with a number they need to be spelled out. It should be 'one'

Author Response

Because there is some formulas in the reply, it could not be shown in this text area. Please see the reply in the supplements.

Reviewer 5 Report

The author studied the influence of temperature on QCM test results. My attitude is negative toward the manuscript.

Temperature affects almost all physical and chemical processes of biosensors, which is a piece of general knowledge. It is impossible to take temperature as a single variable to control the sensor. For example, what is the reason for the temperature influencing SNP detections? Is it necessary to consider the effect of temperature on ion concentration?

Secondly, I question whether the evaporation of aqueous solution causes the ion concentration change?

Then, what do symbols a-k in Figure 2 mean? What is the concentration? This is a terrible problem.

Finally, the mechanism in section 3.5 does not explain any experimental results. Sections 3.1-3.4 are stacked without logic that can be recognized.

Author Response

Because there are some formula in the reply and it could not be pasted in the area, please see the supplement.

Round 2

Reviewer 3 Report

Accept

Reviewer 4 Report

The comments have been addressed

Reviewer 5 Report

The authors have replied to my concerns.

There are no other comments.